# 4-Hydroxycoumarin Exhibits Antinociceptive and Anti-Inflammatory Effects Through Cytokine Modulation: An Integrated In Silico and In Vivo Study

**DOI:** 10.3390/ijms26062788

**Published:** 2025-03-19

**Authors:** Diogo Vilar da Fonsêca, Juliana Sousa Rocha, Pablo R. da Silva, Hugo Natan de Sá Novaes Pereira, Lucas Vinicius Novaes dos Santos, Melquisedec Abiaré Dantas de Santana, Alan F. Alves, Adiel H. O. Pontes, Joás de Souza Gomes, Cícero F. Bezerra Felipe, Damião Pergentino de Sousa, Marcus T. Scotti, Luciana Scotti

**Affiliations:** 1Postgraduate Program in Biosciences, Federal University of the São Francisco Valley (UNIVASF), Pernambuco 56304-917, PE, Brazil; divilar@hotmail.com (D.V.d.F.); julianasrocha0@gmail.com (J.S.R.); 2Postgraduate Program of Dentistry (PPGO), Health Sciences Center, Federal University of Paraíba, João Pessoa 58051-900, PB, Brazil; pablorayff@ltf.ufpb.br; 3Psychopharmacology Laboratory, Institute of Drugs and Medicines Research, Federal University of Paraíba, João Pessoa 58051-900, PB, Brazil; cicero@dbm.ufpb.br; 4Postgraduate Program in Natural Synthetic and Bioactive Products, Health Sciences Center, Federal University of Paraíba, João Pessoa 58051-900, PB, Brazil; 5Department of Medicine, Federal University of the São Francisco Valley (UNIVASF), Paulo Afonso 48605-560, BA, Brazil; hugo.natan@discente.univasf.edu.br; 6Department of Medicine, Faculdade Estácio IDOMED, Juazeiro 48924-999, BA, Brazil; novaess.lucas1@gmail.com; 7Departament of Morphology, Biosciences Center, Federal University of Rio Grande do Norte, Natal 59078-970, RN, Brazil; melquisedec.santana@ufrn.br; 8Cheminformatics Laboratory, Institute of Drugs and Medicines Research, Federal University of Paraíba, João Pessoa 58051-900, PB, Brazil; alvesalanx@gmail.com (A.F.A.); adielsummer99@gmail.com (A.H.O.P.); joasltf@ltf.ufpb.br (J.d.S.G.); mtscotti@gmail.com (M.T.S.); 9Department of Pharmaceutical Sciences, Federal University of Paraíba, João Pessoa 58051-900, PB, Brazil; damiao_desousa@yahoo.com.br

**Keywords:** coumarins, orofacial pain, molecular docking, natural products, multi-target

## Abstract

Chronic pain significantly impacts quality of life and is often accompanied by inflammation, a natural bodily response that can become harmful when excessive. The orofacial region is commonly affected, making effective treatment crucial. However, current drugs often cause undesirable side effects, highlighting the need for new pharmacological alternatives. 4-hydroxycoumarin (4-HC), a natural compound, has shown promising antinociceptive and anti-inflammatory effects, but studies confirming its specific properties are limited. In silico analyses suggest that 4-HC exhibits favorable pharmacokinetic characteristics, not interacting with P-glycoprotein and successfully crossing the blood–brain barrier. Molecular docking studies indicate that its effects may be mediated through NMDA_R_ or by inhibiting iNOS. Our study assessed the antinociceptive and anti-inflammatory effects of 4-HC in animal models at doses of 25, 50, and 75 mg/kg. 4-HC significantly reduced abdominal contortions induced by acetic acid and decreased nociceptive rubbing in orofacial pain models induced by formalin, glutamate, and capsaicin. Interactions with opioid receptors were not observed, suggesting that 4-HC’s antinociceptive effect does not involve this pathway. Additionally, 4-HC reduced paw edema induced by carrageenan and significantly decreased leukocyte migration and TNF-α levels. These findings highlight the therapeutic potential of 4-HC and warrant further investigation into its mechanisms.

## 1. Introduction

Pain is a component of the inflammatory process, and according to the International Association for the Study of Pain, it is defined as “an unpleasant sensory and emotional experience associated with, or similar to, actual or potential tissue damage”. Thus, we see cognitive and emotional components related to the perception and generation of pain, which have not yet been well elucidated [1,2]. Orofacial pain is associated with the tissues of the head, face, neck, and structures of the oral cavity, and may be correlated with genetic, behavioral, psychosocial, cultural, or traumatic factors, or even postural habits. Orofacial pain has significant implications for stomatognathic functions, potentially limiting essential activities such as chewing, swallowing, speaking, and maintaining oral hygiene. These limitations can trigger a cascade of disorders affecting the anatomical and functional structures of the skull and oral cavity and lead to substantial local and systemic impairments for the individual [3].

Inflammation is a physiological reaction of the organism to tissue damage resulting from the presence of a foreign body, trauma (mechanical, chemical, or thermal), infections, immunological reactions, or tissue necrosis. This reaction is related to the release of chemical substances such as cytokines, chemokines (e.g., TNF-α, lipoxins, kinins, prostaglandins, leukotrienes), and proteins that function to signal cell chemotaxis to the damaged tissue, with the aim of blocking, inactivating, or eliminating the causative agent, as well as healing the damaged tissues [4,5].

Inflammation in the orofacial region is one of the main causes of orofacial pain, as it generates the release of pro-inflammatory cytokines, interleukins, and reactive oxygen species in the nerve fibers that make up the region, as well as sensitizing these fibers, causing a decrease in the sensory threshold interpreted as pain [6]. The multifactorial etiology of pain leads to the need for multi-drug, and in most cases, multi-professional intervention [7].

In the context of treatment for both orofacial pain and inflammatory processes, the search for agents that produce fewer adverse effects and that are more effective than those currently marketed is encouraging. Due to the global need to serve those affected by orofacial pain, products of natural origin, such as plants, given their wide availability and extensive biodiversity, being already used in many societies [3,8], deserve further study.

Coumarins, whose molecular structure is based on a common benzo-alpha-pyrone skeleton, are present in the roots, flowers, and fruits of various species of the plant kingdom. Due to their wide biodiversity, they have become important molecules in natural products and medicinal chemistry [9]. They present various properties as elucidated in the literature, including anti-inflammatory [10], antioxidant [11], antiallergic [12], antithrombotic [13], antiviral [14], antitumor [10], and anticoagulant [15] activity.

A representative of the coumarin group, 4-hydroxycoumarin (4-HC) emerges as an interesting alternative due to its excellent therapeutic potential. 4-HC contains a hydroxyl group at the C-4 position, rendering the molecule amphiphilic. It has been studied for various properties, including its insecticidal activity [16], anticancer effects [17], and anti-inflammatory activity in a colitis model [18]. This study sought to better characterize the peripheral/orofacial antinociceptive and anti-inflammatory properties of 4-HC through the use of animal models and in silico simulations aiming to advance our knowledge of its pharmacological properties and improve the treatment of pain and inflammation in orofacial disorders.

## 2. Results

### 2.1. In Silico Tests

The evaluation of pharmacokinetic properties (absorption, distribution, metabolism, excretion) and toxicity of a molecule can help the identification of potential issues that could hinder a compound’s success in later stages, thereby improving the efficiency and success rate of development efforts [19]. According to the model, this molecule acts as a non-inhibitor and non-substrate for the P-glycoprotein and does not inhibit any isoform of the cytochrome p450 complex enzymes (CYPs). In the present model, 4-HC was predicted to be absorbed in human intestinal tissue and be permeable to the blood–brain barrier (BBB). 4-HC also did not present toxicity related to carcinogenesis, suggesting that the molecule probably does not have carcinogenic potential and is safe in the biodegradation model and the Ames test (Table 1). 

Molecular docking is a computational technique employed to predict the optimal orientation of a molecule (ligand) when bound to a specific target (e.g., a protein). This method estimates the binding energy of the formed ligand–target complexes, providing insights into the stability and feasibility of the interaction [20]. Such analyses are crucial for evaluating whether a molecule can modulate a specific biological pathway through its interactions with relevant targets. The molecular docking results, presented in Table 2, summarize the estimated binding energies of the complexes formed between the ligands and the selected targets. Binding energy values are indicative of the stability of the complexes, with more negative values corresponding to more stable interactions [21].

Celecoxib is a highly selective inhibitor of COX-2 compared to COX-1 and is a representative of the coxib class of drugs. It exhibits potent anti-inflammatory effects and has been used as a reference compound in comparative studies [22]. Molecular docking analysis revealed that the complex formed between 4-HC and COX-2 demonstrated lower stability, with a binding score of −54.84, in contrast to the complex formed between COX-2 and celecoxib, which showed significantly higher stability with a score of −158.60. Celecoxib established multiple interactions, including hydrogen bonds, pi–pi stacking, and pi–alkyl interactions, whereas 4-HC formed only pi–alkyl interactions. Furthermore, both complexes interacted with the residues Ala513, Val335, Leu338, and Val509 (Appendix A).

Regarding the GABA_A_ receptor, bicuculline was employed as a comparative reference standard. This molecule functions as a competitive antagonist of the GABA_A_ receptor, exerting its inhibitory effects on GABAergic signaling in key regions involved in pain modulation, such as the spinal cord, periaqueductal gray, and rostral ventromedial medulla [23]. The complex formed between 4-HC and the GABA_A_ receptor exhibited a lower binding affinity with a docking score of −50.64 compared to the complex formed between the receptor and bicuculline with a score of −146.56. The residue Phe200 was the only common interaction site shared by both complexes, highlighting the differences in their binding modes and affinities for the GABAA receptor (Appendix A).

Regarding iNOS, 7-nitroindazole, an antagonist with a high affinity for the iNOS receptor, was used as a comparative standard [6]. The investigation with iNOS revealed that the complex formed between 4-HC and the iNOS protein exhibited a higher binding score (−106.52) compared to the complex formed between iNOS and 7-nitroindazole (−103.34). The residues Gly365 and Tyr367 formed only van der Waals interactions with 4-HC, whereas they established carbon–hydrogen bonds with 7-nitroindazole. The residues Pro344 and Met368 were common to both complexes. Additionally, pi–alkyl interactions were observed with the heme cofactor in both cases (Appendix A).

To investigate whether an intracellular mechanism could be related to an antinociceptive effect, the interaction of 4-HC with NFκB was also analyzed and compared to the corticosteroid dexamethasone. The results showed that NFκB formed a complex with a more favorable binding score with dexamethasone (−91.38) compared to 4-HC (−58.71). Only two amino acid residues were common to both complexes: Asp239 formed a hydrogen bond in both complexes, while His141 exhibited an unfavorable interaction with dexamethasone but formed a hydrophobic pi–alkyl interaction with 4-HC (Appendix A).

To evaluate whether 4-HC could exhibit activity through opioid receptors, the binding energy of 4-HC in complex with the mu-opioid receptor (MOR) was analyzed and compared to morphine. The results showed that the complex formed with morphine achieved a more favorable binding score (−78.23) compared to 4-HC (−28.6). 4-HC formed a single interaction with Met153, along with two van der Waals interactions with Asp149 and Trp295. In contrast, morphine established a carbon–hydrogen bond with Asp149 and a pi–alkyl interaction with Met153 (Appendix A).

Sketamine was employed as a comparative standard to evaluate the binding interactions of 4-HC with NMDA_R_. Sketamine functions as a non-competitive antagonist targeting the GluN2B subunit of NMDA_R_, which is critically involved in pain signaling pathways [24]. The complex formed between 4-HC and NMDA_R_ exhibited a more favorable binding score (−56.12) compared to the complex formed with sketamine (−51.11), suggesting that 4-HC forms a more stable interaction with the receptor. 4-HC established a hydrogen bond with Thr648 and pi–alkyl interactions with Ala647, Ala650, and Ala652. In contrast, sketamine formed hydrogen bonds with Asn614 and Asn615, and pi–alkyl interactions with Leu642 and Val644 (Appendix A).

Finally, capsazepine, a competitive inhibitor of TRPV1, was used as a comparative standard. The complex formed between TRPV1 and 4-HC achieved a docking score of −62.20, while the complex with capsazepine obtained a more favorable score of −87.79. 4-HC formed a hydrogen bond with Glu570 and a pi–alkyl interaction with Leu553, whereas capsazepine exhibited multiple interactions, including pi–alkyl, pi–sulfur, and carbon–hydrogen bonds, alongside unfavorable interactions with Ser512 (Appendix A).

Molecular docking analysis demonstrated the potential interaction of 4-HC with biological targets by predicting its binding affinity, molecular interactions, and conformational stability. The calculated binding energy indicates the strength of the formed complex, while interactions such as hydrogen bonds and hydrophobic forces reinforce the specificity of the binding. Additionally, the analysis identifies the binding site and suggests a mechanism of action, comparing it with known ligands. These data support further experimental studies, validating the therapeutic potential of 4-HC.

### 2.2. In Vivo Tests

#### Effect of 4-HC on the Acetic Acid-Induced Writhing Protocol

The acetic acid-induced writhing test is a widely used in vivo experiment to evaluate the antinociceptive properties of compounds. In this assay, the intraperitoneal injection of acetic acid causes irritation in the peritoneal cavity, leading to the release of inflammatory mediators such as prostaglandins, cytokines, and nitric oxide, which trigger nociceptive responses characterized by abdominal writhing [25]. A reduction in the number of writhing responses in animals treated with 4-HC, compared to the control group, indicates its ability to suppress pain perception, likely through the inhibition of inflammatory mediators. In this study, 4-HC at doses of 25, 50, and 75 mg/kg showed no significant differences between them. However, the 50 mg/kg dose resulted in the greatest reduction (*p* < 0.05) in the number of abdominal contortions, with a decrease of approximately 36.7% compared to the control group. The 25 mg/kg and 75 mg/kg doses produced respective reductions (*p* < 0.05) of 24.4% and 31.7%. In contrast, the positive control group treated with morphine (5 mg/kg) exhibited a complete abolition of acetic acid-induced abdominal contortions. The acetic acid-induced writhing test does not demonstrate dose dependency, as shown in Figure 1. These results reinforce the significant peripheral antinociceptive effects of 4-HC, highlighting its potential as an analgesic and anti-inflammatory agent.

### 2.3. Effect of 4-HC in the Formalin-Induced Orofacial Nociception Protocol

In the first phase, which is in the first 5 min (Figure 2a) of the formalin test, the 4-HC dose of 75 mg/kg (22.9 ± 7.5 s) reduced the time of friction movement with the paws at the applied site in the orofacial region by 60.0% when compared to the control group (57.3 ± 4.4 s). The doses of 25 (50.2 ± 4.4 s) and 50 (44.4 ± 7.5 s) mg/kg did not alter the nociceptive behavior. With 4-HC doses of 50 (48.2 ± 12.2 s) and 75 (45.8 ± 15 s) mg/kg, as demonstrated in the second phase of the protocol (Figure 2b), the animals were observed at 15 to 40 min after the application of formalin, obtaining a reduction in the friction time in the respective percentages of 60.8% and 63.7% when compared to the control group. Morphine reduced the friction movement time with the paws on the face in the first phase by 48.2% and in the second phase by 82.5%.

### 2.4. Effect of 4-HC in the Glutamate-Induced Orofacial Nociception Protocol

In the glutamate-induced orofacial nociception test, 4-HC at doses of 25 (53.17 ± 9.92 s), 50 (67.83 ± 7.14 s), and 75 (54.25 ± 9.41 s) mg/kg promoted a significant reduction in the friction time with the hind or forepaws in the injected area, with respective reduction values of 55.8%, 43.6%, 54.8%, and 69.1% when compared with the control group (Figure 3). Further, the glutamate-induced orofacial nociception protocol was performed and it was observed that 4-HC at doses of 25 mg/kg (53.17 ± 9.92 s), 50 mg/kg (67.83 ± 7.14 s), and 75 mg/kg (54.25 ± 9.41 s) promoted respective significant reductions in the friction time at the injected area of 55.8%, 43.6%, and 54.8% when compared to the control.

#### 2.4.1. Effect of 4-HC on the Capsaicin-Induced Orofacial Nociception Protocol

In the capsaicin test, doses of 50 and 75 mg/kg significantly reduced the time the animal spent rubbing the orofacial region with its front paws in the injected area, with respective reduction values of 63.0% (84.0 ± 11.5 s) and 68.6% (31.0 ± 3.34 s) for 45 min after injection of the substance, as compared with the control (Figure 4).

#### 2.4.2. Opioid Receptors

No significant differences were observed between the groups treated with 4-HC (75 mg/kg) combined with naloxone (2 mg/kg) and those that received only hydroxycou-marin. These results, both in the first phase (0–5 min) and in the second phase (15–30 min), suggest that the mechanism of action of 4-HC does not involve the activation of the opioid system (Figure 5A,B). On the other hand, when comparing the group treated with morphine to the morphine + naloxone group, a reversal of the effect is observed, which confirms naloxone’s action and reinforces the mechanism of action of the standard employed in the experiment.

### 2.5. Carrageenan-Induced Paw Edema Test

The treatment performed with 4-HC at a dose of 75 mg/kg presented a reduction in all test times (60, 120, 180, and 240 min) and significantly reduced paw edema in the following respective proportions: 72.0%, 70.6%, 74.6%, and 66.3%. This was compared to the control and with the group treated with dexamethasone, which was used as a positive control. The 50 mg/kg dose of 4-HC reduced paw edema by 52.8% within 120 min of testing when compared to the control, as shown in Figure 6.

### 2.6. Leukocyte Count Test and Pro-Inflammatory Cytokine TNF-α Dosage

To analyze the anti-inflammatory action of 4-hydroxycoumarin, animals treated with carrageenan via i.p. were subjected to leukocyte counts. As observed in Figure 7 animals treated with 4-HC 75 mg/kg (4.8 ± 0.8 × 10^5^ /mL) demonstrated a significant decrease in leukocyte migration compared to the control group (12.4 ± 0.7 × 10^5^ /mL) (*p* < 0.001). Animals that received dexamethasone 2 mg/kg demonstrated a mean leukocyte count of 6.3 ± 0.6 × 10^5^ /mL (*p* < 0.001) (Figure 7).

The pro-inflammatory cytokine TNF-α was measured in the peritoneal fluid, in which a significant reduction of 71.8% was observed in the group that received 4-HC 75 mg/kg (110.6 ± 8.2) in relation to the control group (392.3 ± 86.0), with a *p*-value < 0.01. The group that received dexamethasone (2 mg/kg) presented a dosage of 103.4 ± 4.3, as shown in Figure 8.

## 3. Discussion

According to pharmacokinetic predictions, 4-HC exhibited favorable properties. The lack of interaction with P-glycoprotein suggests an enhanced safety profile, as this protein functions as an efflux pump, transporting numerous drugs out of the cells. Interaction with P-glycoprotein could increase intracellular concentrations of the substance, raising the likelihood of side effects and adverse outcomes [26]. Furthermore, the non-inhibition of CYP P450 enzymes and P-glycoprotein indicates that 4-HC is likely safe and may cause minimal or no drug–drug interactions. CYP enzymes, a subfamily of oxidases, play a central role in the first-pass metabolism of most marketed drugs, and their inhibition can lead to drug interactions and increased adverse effects [27]. Such inhibition could result in changes in pharmacodynamics and elevated toxicological risks [28]. Additionally, 4-HC demonstrated safety in terms of mutagenicity, as evidenced by the Ames test, and carcinogenicity, further supporting its pharmacokinetic safety profile.

The molecular docking analysis revealed no significant interactions with the GABAA, NFκB, µ-opioid [29], or TRPV receptors, suggesting that these proteins may not be directly involved in the mechanism of action. However, the effects may be synergistic, making in vivo investigation of these targets necessary. In contrast, the complex formed with iNOS exhibited a binding score very close to that of the reference ligand, indicating a potential interaction at this site. Furthermore, the complex with NMDAR demonstrated a higher binding score compared to that of S-ketamine, suggesting a possible inhibitory effect on the NMDAR receptor. These findings imply that the compound may exert its primary effects through interactions with iNOS and NMDAR, rather than the other tested targets. The selection of these molecular targets was based on the current literature regarding the mechanisms underlying pain, which involve the activation of various biochemical pathways, including the induction of nitric oxide by iNOS and signaling mediated by NMDAR receptors. The activation of NMDAR receptors allows for calcium to enter neuronal cells, which can trigger central sensitization and amplify the transmission of harmful signals. When these receptors are blocked, calcium entry is inhibited, reducing the response to the painful stimulus. In the case of iNOS, its increased expression is associated with heightened inflammation, as the nitric oxide (NO) generated can activate C-fiber nociceptors and contribute to the onset of painful sensations in the spinal cord [30,31,32].

4-HC presented positive results for the models performed, evidencing an orofacial antinociceptive and anti-inflammatory effect. In this sense, to verify these properties, tests were performed inducing orofacial pain and evaluated through specific motor responses. When times were reduced in the groups that received 4-HC, antinociceptive activity was considered present according to the response time of each group. Similarly, the anti-inflammatory response of 4-HC was evaluated through the study of components present in the inflammatory process in the animal model, such as the evaluation of edema size and the presence of leukocytes and pro-inflammatory cells in the stimulated regions.

To evaluate the nociceptive properties of 4-HC, the abdominal contortions test induced by acetic acid was conducted. This is a highly sensitive, non-selective experimental model for testing anti-inflammatory substances. The test involves the activation of visceral somatic receptors and local inflammatory processes mediated by prostaglandins, bradykinins, and inflammatory cytokines such as TNF-α, IL-6, IL-1β, and IL-8 [33]. A reduction in the number of contortions was observed in the acetic acid test with the use of 4-HC, indicating its potential antinociceptive effect, which may act through the reduction of these pro-inflammatory cytokines, as was characterized for other coumarins using the same test, such as bergapten [34] and imperatorin

The orofacial nociception induced by formalin protocol was performed, in which it is possible to demonstrate two different types of pain at different times, demonstrating the analgesic effect of different drugs with different mechanisms of action [35]. It was observed that 4-HC presented an orofacial antinociceptive profile in the formalin test in the first phase of the test, which was characterized by the direct stimulation of type C afferent nociceptors. This result may signify that the mechanism of action is a reduction in the release of inflammatory mediators, or direct blockage of these receptors. In the study carried out by Gripp et al., 2020 [32], a coumarin fraction was extracted from the leaves and stems of *Anaxagorea dolichocarpa* to perform the formalin test, in which antinociceptive activity in the animal’s paw was evidenced, as there was a reduction in the act of paw licking compared to the control group.

The second phase of the formalin test (between 15 and 30 min) is related to the release of pro-inflammatory mediators such as bradykinin, histamine, substance P, serotonin, and prostaglandins, which interact with their respective receptors to manifest inflammatory pain. In this stage of the test, 4-HC presented satisfactory results, which corroborates a study by Cheriyan et al., 2017 [33], where 7-methoxycoumarin also obtained a significant response in the second phase of nociception. In addition, the coumarin imperatorin presents an inhibitory effect on formalin-induced pain in phase II via the inhibition of Nav1.7 [36]. We, therefore, suggest that 4-HC may have a central antinociceptive effect for obtaining favorable results in both phases of the formalin test.

Further, once administered in the orofacial region, 4-HC causes the activation of receptors in peripheral, spinal, and supraspinal sites by a mechanism dependent on the activation of the L-citrulline and the nitric oxide pathway. Glutamate-induced nociception is related to N-methyl D-aspartate (NMDAR) and non-NMDA receptors located in the spinal, supraspinal, and peripheral regions [37]. In this sense, a significant reduction in the friction time of the front or rear paws of mice was observed at all doses in the glutamate test with 4-HC, which indicates the likely participation of this pathway in the antinociceptive effect of 4-HC.

The capsaicin protocol revealed significant antinociceptive effects for 4-HC at doses of 50 and 75 mg/kg, as there was a reduction in the time of friction at the mouse face. It was therefore observed that 4-HC presents antinociceptive activity, acting at a central level since capsaicin injection increases excitability in spinal nociceptive neurons and is widely used in the preclinical study of the central mechanisms involved in nociception [38]. The action of opioids occurs through the interaction of agonists or antagonists, whether synthetic, endogenous, or natural. These bind to opioid receptors, classified as µ, κ, δ, ζ, ε, σ, and ORL-1 [39].

The pretreatment with naloxone did not antagonize the antinociceptive effect of 4-HC, characterized by the reduction in paw friction, suggesting that the opioid system is not involved in this mechanism. Naloxone, a nonspecific opioid receptor antagonist with a high affinity for µ-opioid receptors [40], acts rapidly; however, its inability to reverse the effect of 4-HC indicates that opioid receptors do not play a significant role in mediating this action. These results are consistent with the in silico study, which showed no affinity (binding and satisfactory energy) for opioid receptors. Together, these findings further support the conclusion that the antinociceptive effects of 4-HC are likely mediated through non-opioid pathways, reinforcing its potential as a novel analgesic with a distinct mechanism of action.

To corroborate the potential anti-inflammatory activity demonstrated in the second phase of the formalin test, the carrageenan-induced paw edema model was evaluated. Carrageenan is an inflammatory agent responsible for triggering inflammatory processes through the release of prostaglandins and causing edema [41]. Recent studies have characterized the edema induced by carrageenan in an initial phase (1–6 h) as being the phase related to the increase in the levels of prostaglandins and thromboxane, accompanied by the attenuated expression of the enzyme cyclooxygenase-2 (COX-2) [42].

In this sense, treatment with 4-HC at a dose of 75 mg/kg significantly reduced carrageenan-induced paw edema at all test times (60, 120, 180, and 240 min). This is corroborated by studies carried out with other coumarins, such as 3-coumarin carboxylic acid, which has been shown to exert its effect peripherally, exhibiting anti-inflammatory properties to reduce carrageenan-induced paw edema [8] (Aragão, 2022). To evaluate the anti-inflammatory properties of 4-HC, leukocyte count and TNF-α levels were tested by analyzing the peritoneal fluid of mice that had previously undergone peritonitis induction with 1% carrageenan. The group of animals receiving 4-HC 75 mg/kg before the pro-inflammatory stimulus presented a significant reduction in both leukocyte count and TNF-α levels compared to the control group, which indicated its anti-inflammatory capacity in this experiment. The mechanism by which 4-HC lowers TNF-α levels is still undefined. It is proposed that the compound may act through the inhibition of the NFκB pathway via downregulation of pro-inflammatory cytokines. Therefore, additional experimental assays are needed to confirm which pathway is involved in this effect.

The significant reduction in leukocyte count has been previously described in the literature for certain hydroxyl-substituted aromatic coumarin derivatives, such as 5-hydroxycoumarin or vicinal dihydroxycoumarins, considered to be potent anti-inflammatory agents, as their amphiphilic capacity allows for greater tissue permeability [15]. It has also been shown that coumarins exert anti-inflammatory activity. This is related to leukocyte migration decreases. This is the case with esculetin, a coumarin that exerts this effect by reducing the level of adhesion to endothelial cells, which inhibits the secretion of soluble intercellular adhesion molecules responsible for inflammation [43].

As for the significant reduction in the TNF-α dosage, anti-inflammatory activity is common to this family of substances. This has been observed in other studies involving coumarins and flavones and may occur through the inhibition of NFkB activation by preventing the phosphorylation and degradation of IκBα [18]. Furthermore, it has been reported in the literature that scopoletin and esculin have anti-inflammatory effects related to the reduction in the expression levels of pro-inflammatory factors such as TNF-α, but also IL-1ß and IL-6, in addition to expressing several endogenous antioxidant proteins, such as superoxide dismutase, glutathione peroxidase, and glutathione reductase [44,45]. All results together demonstrate the important anti-inflammatory and antinociceptive effect of the substance 4-HC.

Therefore, despite the promising initial results, more research is needed to elucidate the pharmacodynamics of 4-HC, as its specific mechanisms of action at the molecular and cellular levels are not fully understood. While pharmacokinetic studies may indicate potential therapeutic benefits, it is crucial to investigate how 4-HC interacts with its biological targets (such as receptors, enzymes, or signaling pathways) to determine its efficacy and safety. Additionally, understanding the dose–response relationship, therapeutic window, and potential side effects will help optimize its therapeutic use. However, this will be addressed in future studies.

## 4. Materials and Methods

### 4.1. Computational Studies

Pharmacokinetic and toxicity analyses were performed using the Deep-PK web tool (https://biosig.lab.uq.edu.au/deeppk) (accessed on 12 December 2024) by inputting the .sdf file containing the chemical structure of cannabidiol and selecting the option “ADMET” [46]. 4-HC was modeled using MarvinSketch software v.23.14 and optimized using the semi-empirical MMFF method of Spartan v.14 [13]. The crystallographic structures of six targets were selected from the Protein Data Bank (PDB) (Table 3) based on their involvement with pain physiopathology. Antagonists (COX-2, GABAA, iNOS, NFκB, and NMDAR) or agonists (µ-opioid receptor) of each target were selected for comparison. The amino acid sequence utilized for NFκB was detailed in the work of Piccagli et al. [47].

The Molegro Virtual Docker software v.2013.6.0.1 [22] was used to perform molecular docking. All water molecules and cofactors were excluded, except for the group heme in iNOS, which interacts with the bounding site. Before molecular docking, the redocking step was performed to evaluate the accuracy and reliability of the results from the Root–Mean–Square Deviation (RMSD). The RMSD is a necessary step to verify that the algorithm was able to produce the correct pose, and values ≤ 2 Å are considered satisfactory. RMSD values were not considered for the NMDA receptor (PDB: 7EOQ) as it did not have a co-crystallized ligand available, and sketamine was used as a positive control.

The simulation was conducted using the default settings. The MolDock Score function was used to evaluate the ligand poses, considering internal energy, hydrogen bonds, and torsional energy contributions. Twenty runs were performed using the MolDock SE algorithm, and the top five poses were retained. A grid with a radius of 15 Å and a resolution of 0.30 Å was generated, centered on the positions of the crystallographic ligands on the selected proteins. The fitted poses were further analyzed using Discovery Studio Visualizer v21.1.0.20298 [23].

### 4.2. In Vivo Tests

#### 4.2.1. Animals

Male Swiss mice (Mus Musculus), weighing 25 to 35 g, 3 months old, from the Central Animal Facility of the Universidade Federal do Vale do São Francisco (UNIVASF), Pernambuco, Northeastern Brazil, were kept in a 12 h light–dark cycle under controlled temperature conditions (21 ± 1 °C), with distilled water and pellet food freely available. All experimental procedures were analyzed and previously approved by the Animal Research Ethics Committee (CEPA) of UNIVASF under certificate No. 0002/250221.

#### 4.2.2. Materials Preparation

4-hydroxycoumarin (4-HC) was purchased from Sigma-Aldrich Brazil LTDA. (São Paulo, Brazil). TNF-α was obtained from eBioscience (San Diego, CA, USA). Dexamethasone, capsaicin, and carrageenan were purchased from Sigma (St. Louis, MO, USA). All drugs were diluted in distilled water, except for 4-HC, which was dissolved in Tween 80 and distilled water. All doses of the substances were administered via the i.p. route, except for formalin, glutamate, and capsaicin, which were administered via the orofacial subcutaneous route. Carrageenan was administered via the i.p. route in the animals’ paws.

#### 4.2.3. Acetic Acid-Induced Abdominal Contortions Test

The acetic acid-induced abdominal writhing test consists of the intraperitoneal administration of a 1% acetic acid solution, causing peritoneal irritation that promotes stimulation of nociceptors and release of inflammatory mediators, generating behavioral reactions [25]. In this context, the animals were divided into groups (n = 8/group) to receive the following treatments: vehicle, 4-hydroxycoumarin (25, 50, and 75 mg/kg, i.p.), or morphine (5 mg/kg, i.p.). Thirty minutes after the final administration, a 1% acetic acid solution was injected intraperitoneally, and then the mice were individually placed in polyethylene boxes. The number of abdominal writhings was then recorded for 10 min. Antinociceptive activity was defined as a significant inhibition of the number of abdominal writhings of the animals treated with the test drugs when compared to the control group.

#### 4.2.4. Formalin-Induced Orofacial Nociception Test

For this test, mice were separated into five groups (n = 8/group) treated with vehicle, 4-hydroxycoumarin (25, 50, and 75 mg/kg, i.p.), or morphine (5 mg/kg, i.p.). Trigeminal nociception was assessed by administering 20 µL of a 2% formalin solution into the orofacial region of the mouse, which induces stimulation of nociceptors The behavioral responses observed in this experimental model were assessed using the self-cleaning (grooming) time directed at the formalin injection site [55]. The first phase of the test usually occurs within the first 5 min after formalin injection and is likely the result of direct stimulation of nociceptors, leading to a neurogenic response. This is followed by an interaction period lasting approximately 10 min, identified by pain inhibitory mechanisms. The second phase (15–40 min) is known primarily as an inflammatory response, generated by both nociceptor stimulation and the release of inflammatory mediators [56].

#### 4.2.5. Glutamate-Induced Orofacial Nociception Test

Glutamate is an excitatory neurotransmitter and acts by rapid transmission of painful stimuli through binding to the mGluR receptor. The activation of these receptors promotes a voltage-dependent influx of calcium and sodium and the propagation of neuronal excitability The glutamate-induced nociception test was initially described by Beirith et al., 2002 [37], and later adapted for the orofacial region. The test consisted of subcutaneous administration of 40 µL of a 25 µM glutamate solution (Sigma Aldrich, St. Louis, MO, USA) into the right upper lip of mice treated with vehicle; doses of 25, 50, and 75 mg/kg of 4-HC intraperitoneally; or morphine (5 mg/kg, i.p.). After glutamate administration, the animals were placed in an observation box and monitored for their orofacial nociceptive behavior for 15 min [35]

#### 4.2.6. Capsaicin-Induced Orofacial Nociception Test

Chemical nociception induced by orofacial injection of capsaicin in mice followed the experimental model proposed by Pelissier et al., 2002 [57] with some modifications. The animals were divided into groups that received 4-hydroxycoumarin (50 and 75 mg/kg, i.p.) or morphine (5 mg/kg, i.p.) as a positive control, and the negative control group was treated only with the vehicle (10 mL/kg, i.p.). After 30 min, 20 µL of capsaicin (1.6 µg or 5.2 nmol) was administered to the mice in the right upper lip near the vibrissae region. Immediately after the injection, the animals were placed under the bench with mirrors positioned at a 45° angle to facilitate observation. The mice were observed for 45 min, and the time during which the animals remained rubbing the orofacial region with their front paws was timed and considered indicative of nociception [57].

#### 4.2.7. Investigation of the Opioid System in the Orofacial Antinociceptive Activity of 4-Hydroxycoumarin

To evaluate the participation of µ and κ-type opioid receptors in the antinociceptive activity of 4-HC, a group of animals was treated with a non-selective opioid antagonist (naloxone, 2 mg/kg, i.p.), fifteen minutes before the administration of morphine (5 mg/kg, i.p.) or 4-HC (75 mg/kg, i.p.). At 30 min after the administration of morphine and 60 min after treatment with 4-HC, the animals were subjected to the formalin test.

#### 4.2.8. Evaluation of Anti-Inflammatory Activity Using the Carrageenan-Induced Paw Edema Protocol

Swiss mice, divided into groups of four, were treated with vehicle (80% Tween solution) or 4-hydroxycoumarin (50 and 75 mg/kg i.p.). Sixty minutes after administration, the animals received an injection of 1% carrageenan (i.p.) to induce edema in the right hind paw. The paw volume was recorded in triplicate and the mean was considered before and after (1, 2, 3, and 4 h) the administration of carrageenan. The edema volume in milliliters (mL) was recorded through a plethysmograph where the paw was submerged up to the tibiotarsal junction in the reading chamber of the device. The volume of the displaced liquid was recorded and considered the paw volume. The results were expressed as the difference between the paw volume at the mentioned intervals and the volume before the injection of carrageenan.

#### 4.2.9. Leukocyte Count Test and TNF-α Cytokine Dosage

In this experiment, test groups (n = 7 per group) were treated with 4-HC (75 mg/kg, i.p.), dexamethasone (2 mg/kg, s.c.), or vehicle (Tween 80 solution). Additionally, a control group received only 0.9% sodium chloride without peritonitis induction. After 30 min of pretreatment, the animals were subjected to peritonitis induction with 1% carrageenan in the peritoneal cavity. After four hours, the animals were euthanized and the peritoneal fluid was collected for total leukocyte count and TNF-α cytokine dosage using the Elisa technique.

### 4.3. Statistical Analysis

The statistical tests used were defined according to the characteristics of the experiment and were analyzed using ANOVA followed by Dunnett’s test (for parametric measurements). The values obtained were expressed as mean ± standard error of the mean (SEM), and the results were considered significant when *p* < 0.05.

## 5. Conclusions

The results in silico show that 4-HC has good pharmacokinetic parameters, not interacting with P-glycoprotein and crossing the blood–brain barrier, while molecular docking indicates that the effects are likely via NMDAR or by inhibiting iNOS. The murine protocols performed demonstrated that 4-HC presents orofacial antinociceptive activity with no effect on GABA_A_ receptors or on the µ and κ opioid receptors. Further, our study also demonstrated the anti-inflammatory capacity of 4-HC through its ability to reduce carrageenan-induced paw edema, reduce leukocyte counts, and reduce TNF-α in peritoneal fluid. However, further studies are needed to identify additional mechanisms involved in the pharmacodynamics of 4-HC, along with any other additional beneficial activities of this compound.

## Figures and Tables

**Figure 1 ijms-26-02788-f001:**
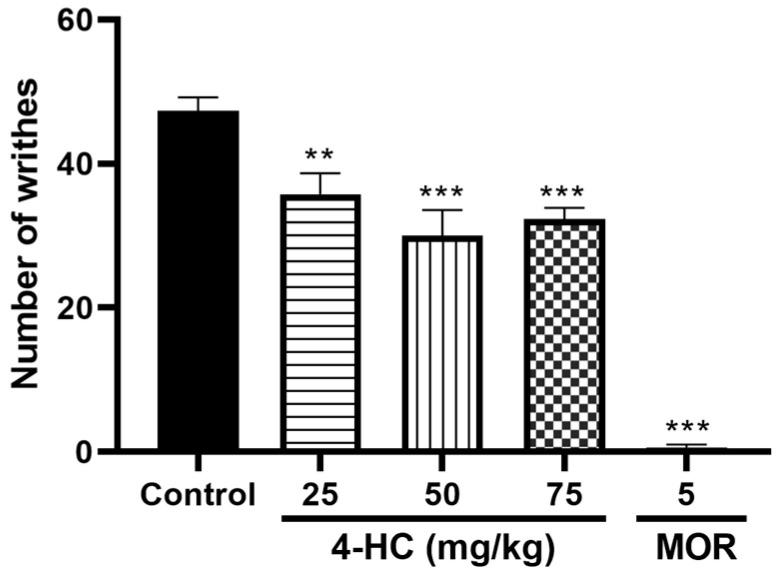
Effect of 4-HC (25, 50, and 75 mg/kg, i.p.) and morphine (MOR: 5 mg/kg, i.p.) on the number of abdominal writhings induced by acetic acid. Each column represents mean ± SEM (n = 8); ** (*p* < 0.01) and *** (*p* < 0.001) (compared with the control group: one-way ANOVA, with Dunnett’s post-test).

**Figure 2 ijms-26-02788-f002:**
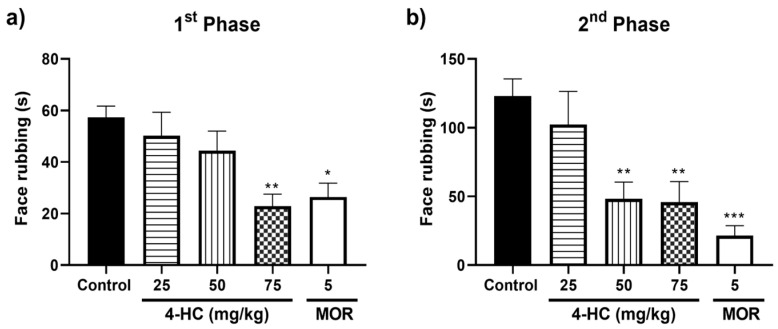
Effect of 4-HC (25, 50, and 75 mg/kg, i.p.) and morphine (MOR: 5 mg/kg, i.p.) in the first (**a**) and second phases (**b**) of the formalin protocol. Each column represents mean ± SEM (n = 8); * (*p* ˂ 0.05), ** (*p* < 0.01) and *** (*p* < 0.001) (compared with the control group: one-way ANOVA, with Dunnett’s post-test).

**Figure 3 ijms-26-02788-f003:**
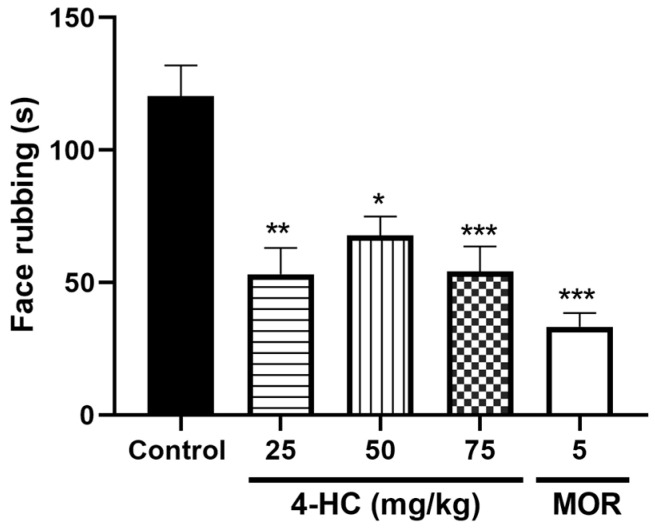
Effect of 4-HC (25, 50, and 75 mg/kg, i.p.) and morphine (MOR: 5 mg/kg, i.p.) in the orofacial glutamate protocol. Each column represents mean ±SEM (n = 8); * (*p* ˂ 0.05), ** (*p* < 0.01) and *** (*p* < 0.001) (compared with the control group: one-way ANOVA, with Dunnett’s post-test).

**Figure 4 ijms-26-02788-f004:**
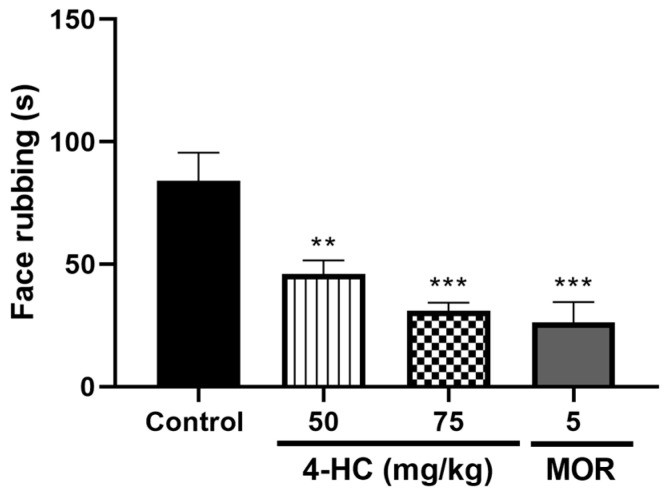
Effect of 4-HC (50 and 75 mg/kg, i.p.) and morphine (MOR: 5 mg/kg, i.p.) in the capsaicin protocol. Each column represents mean ±SEM (n = 8); ** (*p* < 0.01) and *** (*p* < 0.001) (compared with the control group: one-way ANOVA, with Dunnett’s post-test).

**Figure 5 ijms-26-02788-f005:**
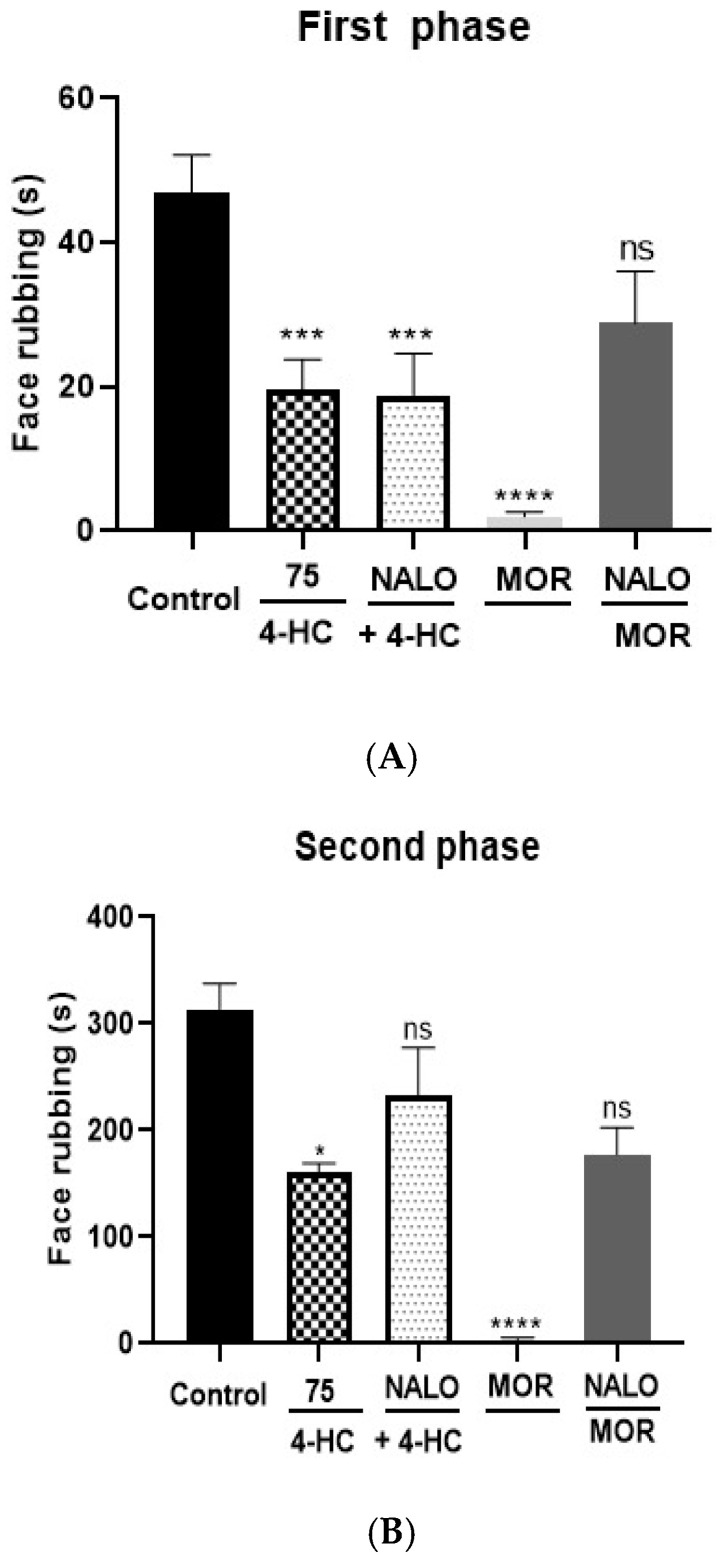
(**A**,**B**) Effect of pretreatment with naloxone (5 mg/kg, i.p.) on antinociception caused by 4-HC (75 mg/kg, i.p.) in the first (**A**) and second (**B**) phases of the formalin protocol. Each column represents mean ±SEM (n = 8); * (*p* ˂ 0.05), *** (*p* < 0.001) and **** (*p* < 0.0001) (compared with the control group: one-way ANOVA, with Dunnett’s post-test). Legend: ns: not significant.

**Figure 6 ijms-26-02788-f006:**
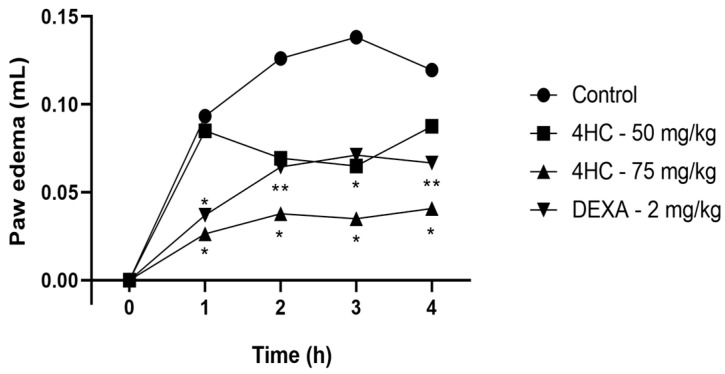
Effect of 4-HC (50 and 75 mg/kg) and dexamethasone (2 mg/kg) in the carrageenan-induced paw edema protocol at 60, 120, 180, and 240 min. Each column represents mean ± SEM (n = 8); * (*p* ˂ 0.05) and ** (*p* < 0.01) (compared with the control group: one-way ANOVA, with Dunnett’s post-test).

**Figure 7 ijms-26-02788-f007:**
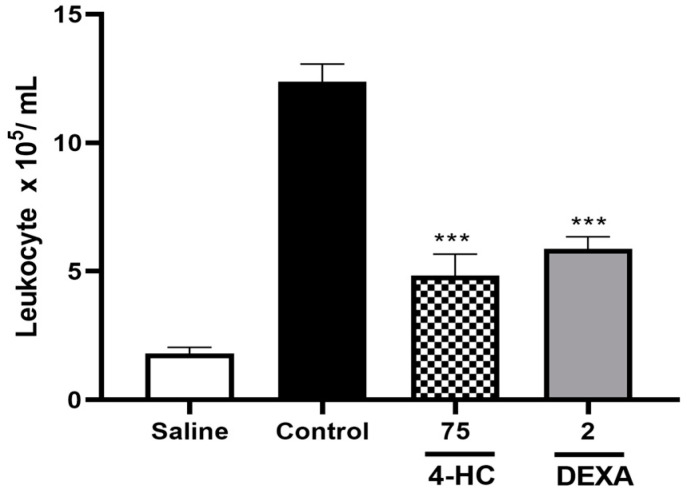
Effect of 4-hydroxycoumarin 75 mg/kg and dexamethasone 2 mg/kg on leukocyte count × 10^5^ /mL. Each column represents mean ± SEM (n = 7). *** *p* < 0.001, versus control group.

**Figure 8 ijms-26-02788-f008:**
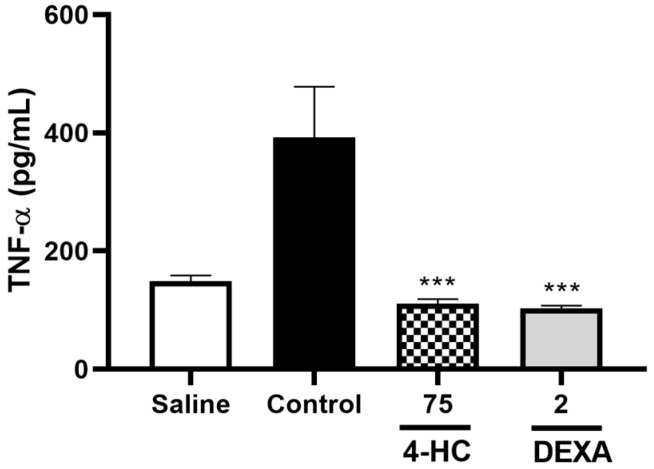
Effect of 4-hydrocoumarin 75 mg/kg and dexamethasone 2 mg/kg on TNF-α pg/mL dosage. Each column represents mean ± SEM (n = 7). *** *p* < 0.001, versus control group.

**Table 1 ijms-26-02788-t001:** Pharmacokinetic and toxicity properties of 4-HC calculated by Deep-PK.

Absorption	
P-Glycoprotein Inhibitor	Non-Inhibitor
P-Glycoprotein Substrate	Non-Substrate
Human Intestinal Absorption	Absorbed
**Distribution**	
BBB	Penetrable
**Excretion**	
Clearance	5.76
Half-Life of Drug	Half-Life < 3 h
**Metabolism**	
CYP 1A2 Inhibitor/Substrate	Non-Inhibitor/Substrate
CYP 2C19 Inhibitor/Substrate	Non-Inhibitor/Non-Substrate
CYP 2C9 Inhibitor/Substrate	Non-Inhibitor/Non-Substrate
CYP 2D6 Inhibitor/Substrate	Non-Inhibitor/Substrate
CYP 3A4 Inhibitor/Substrate	Non-Inhibitor/Non-Substrate
**Toxicity**	
Biodegradation	Safe
Carcinogenesis	Safe
Ames Mutagenesis	Safe

**Table 2 ijms-26-02788-t002:** MolDock scores and RMSD for different targets and ligands.

Target	Ligands	Escore MolDock	RMSD
COX-2	4-HC	−54.84	0.27
	Celecoxib	−158.60	
GABA_A_	4-HC	−50.64	0.08
	Bicuculline	−146.56	
iNOS	4-HC	−106.52	0.19
	7-Nitroindazole	−103.34	
NFκB	4-HC	−58.71	-
	Dexamethasone	−91.38	
NMDA_R_	4-HC	−56.12	-
	Sketamine	−51.11	
µ-opioid receptor	4-HC	−28.36	0.11
	Morphine	−78.23	
TRPV1	4-HC	−62.20	0.66
	Capsazepine	−87.79	

**Table 3 ijms-26-02788-t003:** Targets and ligands used in molecular docking.

Target	PDB (ID)	Resolution	Ligand
COX-2	3LN1 [48]	2.40 Å	Celecoxib
GABAA	6X3S [49]	3.12 Å	Bicuculline
iNOS	1M8E [50]	2.90 Å	7-Nitroindazole
NFκB	1NFK [51]	2.30 Å	Dexamethasone
NMDAR	7EOQ [52]	3.50 Å	Sketamine
µ-opioid receptor	8EF6 [53]	3.20 Å	Morphine
TRPV1	5IS0 [54]	3.43 Å	Capsazepine

## Data Availability

The data presented in this study are available on request from the corresponding author.

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
