# Peer review of "4-Hydroxycoumarin Exhibits Antinociceptive and Anti-Inflammatory Effects Through Cytokine Modulation: An Integrated In Silico and In Vivo Study"

_ijms, 2025, doi:10.3390/ijms26062788_

Round 1
Reviewer 1 Report
Comments and Suggestions for Authors
Minor comments:
Line 175–181: How does the molecular docking analysis confirm the potential interaction of 4-HC with specific biological targets?
Line 163–167: What is the significance of the observed binding affinity of 4-HC with NMDAR compared to sketamine?
Line 184–191: How does the acetic acid-induced writhing test validate the antinociceptive properties of 4-HC?
Line 356–360: What evidence suggests that the opioid system is not involved in the antinociceptive effect of 4-HC?
Line 362–369: How does the carrageenan-induced paw edema model support the anti-inflammatory activity of 4-HC?
Line 387–391: What mechanisms are proposed for the reduction of TNF-α levels after 4-HC treatment? Line 94–103: What pharmacokinetic properties of 4-HC suggest its potential as a therapeutic agent?
Line 530–531: Why is further research needed to elucidate the pharmacodynamics of 4-HC?
Author Response
Title:
Dear Editor,
We appreciated the review of our manuscript and the suggestions given to make it suitable to International Journal of Molecular Science high quality standards. As requested, this letter responds to each point raised by the reviewers. Please, note that changes were tracked-up in the manuscript and are transcribed here:
#Reviewer 1:
Minor comments:
- Line 175–181: How does the molecular docking analysis confirm the potential interaction of 4-HC with specific biological targets?
- Corrected as suggested: Molecular docking analysis demonstrated the potential interaction of 4-HC with biological targets by predicting its binding affinity, molecular interactions, and conformational stability. The calculated binding energy indicates the strength of the formed complex, while interactions such as hydrogen bonds and hydrophobic forces reinforce the specificity of the binding. Additionally, the analysis identifies the binding site and suggests a mechanism of action, comparing it with known ligands. These data support further experimental studies, validating the therapeutic potential of 4-HC.
- Line 163–167: What is the significance of the observed binding affinity of 4-HC with NMDAR compared to sketamine?
- Corrected as suggested:
The observed binding affinity of 4-HC with NMDAR suggests that 4-HC forms a more stable complex with this protein compared to sketamine. Once NMDAR is involved in pain signaling, its inhibition could lead to pharmacological implications, disrupting this pathway.
- Line 184–191: How does the acetic acid-induced writhing test validate the antinociceptive properties of 4-HC?
- Corrected as suggested: The acetic acid-induced writhing test is a widely used in vivo experiment to evaluate the antinociceptive properties of compounds. In this assay, intraperitoneal injection of acetic acid causes irritation in the peritoneal cavity, leading to the release of inflammatory mediators such as prostaglandins, cytokines, and nitric oxide, which trigger nociceptive responses characterized by abdominal writhing. A reduction in the number of writhing responses in animals treated with 4-HC, compared to the control group, indicates its ability to suppress pain perception, likely through the inhibition of inflammatory mediators. In this study, 4-HC at doses of 25, 50, and 75 mg/kg showed no significant differences between them. However, the 50 mg/kg dose resulted in the greatest reduction (p<0.05) in the number of abdominal contortions, with a decrease of approximately 36.7% compared to the control group. The 25 mg/kg and 75 mg/kg doses produced respective reductions (p<0.05) of 24.4% and 31.7%. In contrast, the positive control group treated with morphine (5 mg/kg) exhibited a complete abolition of acetic acid-induced abdominal contortions, as shown in Figure 1. These results reinforce the significant peripheral antinociceptive effects of 4-HC, highlighting its potential as an analgesic and anti-inflammatory agent.
- Line 356–360: What evidence suggests that the opioid system is not involved in the antinociceptive effect of 4-HC?
- Corrected as suggested: The pretreatment with naloxone did not antagonize the antinociceptive effect of 4-HC, characterized by the reduction in paw friction, suggesting that the opioid system is not involved in this mechanism. Naloxone, a nonspecific opioid receptor antagonist with high affinity for µ-opioid receptors, acts rapidly; however, its inability to reverse the effect of 4-HC indicates that opioid receptors do not play a significant role in mediating this action. These results are consistent with the in silico study, which showed no affinity (binding and satisfactory energy) for opioid receptors. Together, these findings further support the conclusion that the antinociceptive effects of 4-HC are likely mediated through non-opioid pathways, reinforcing its potential as a novel analgesic with a distinct mechanism of action.
- Line 362–369: How does the carrageenan-induced paw edema model support the anti-inflammatory activity of 4-HC?
- Corrected as suggested: The carrageenan-induced paw edema model is a widely used in vivo test to evaluate anti-inflammatory activity. In this model, the injection of carrageenan into the paw of a rat or mouse induces acute inflammation, characterized by swelling due to the release of pro-inflammatory mediators, such as prostaglandins, cytokines, and histamines. The increase in paw volume serves as an indicator of the inflammatory response. The anti-inflammatory activity of 4-HC is supported by this model if treatment with the compound leads to a significant reduction in the paw edema when compared to the control group. This reduction suggests that 4-HC is able to modulate the inflammatory response, likely by inhibiting the production or activity of pro-inflammatory mediators involved in the edema formation. A significant reduction in paw swelling after 4-HC treatment implies its potential to reduce acute inflammation, which supports its role as an anti-inflammatory agent.
- Line 387–391: What mechanisms are proposed for the reduction of TNF-α levels after 4-HC treatment? Line 94–103: What pharmacokinetic properties of 4-HC suggest its potential as a therapeutic agent?
- Corrected as suggested: The mechanism by which 4-HC lowers TNF-α levels is still undefined. It is proposed that the compound may act through the inhibition of the NFκB pathway via downregulation of pro-inflammatory cytokines. Therefore, additional experimental assays are needed to confirm which pathway is involved in this effect.
- Corrected as suggested: The pharmacokinetic properties of 4-HC, determined through in silico studies, suggest its potential as a therapeutic agent, including good absorption, effective distribution in target tissues, and adequate metabolism and elimination, with no risk of accumulation. Additionally, its low toxicity and wide therapeutic index support its feasibility for preclinical development.
According to pharmacokinetic predictions, 4-HC exhibited favorable properties. The lack of interaction with P-glycoprotein suggests an enhanced safety profile, as this protein functions as an efflux pump, transporting numerous drugs out of cells. Interaction with P-glycoprotein could increase intracellular concentrations of the substance, raising the likelihood of side effects and adverse outcomes [26]. Furthermore, the non-inhibition of CYP P450 enzymes and P-glycoprotein indicates that 4-HC is likely safe and may cause minimal or no drug-drug interactions. CYP enzymes, a subfamily of oxidases, play a central role in the first-pass metabolism of most marketed drugs, and their inhibition can lead to drug interactions and increased adverse effects [27]. Such inhibition could result in changes in pharmacodynamics and elevated toxicological risks [28]. Additionally, 4-HC demonstrated safety in terms of mutagenicity, as evidenced by the Ames test, and carcinogenicity, further supporting its pharmacokinetic safety profile.
- Line 530–531: Why is further research needed to elucidate the pharmacodynamics of 4-HC?
- Corrected as suggested: Therefore, despite the promising initial results, more research is needed to elucidate the pharmacodynamics of 4-HC, as its specific mechanisms of action at the molecular and cellular levels are not fully understood. While pharmacokinetic studies may indicate potential therapeutic benefits, it is crucial to investigate how 4-HC interacts with its biological targets (such as receptors, enzymes, or signaling pathways) to determine its efficacy and safety. Additionally, understanding the dose-response relationship, therapeutic window, and potential side effects will help optimize its therapeutic use. However, this will be addressed in future studies.
#Reviewer 2:
This manuscript focuses on the in silico and in vivo experiments of 4-hydroxycoumarin. While I appreciate the authors' efforts in preparing the manuscript, several major concerns must be addressed before it can be considered for publication.
- Introduction (Line 79) The term "bioavailability" does not mean "available in biology." It refers to the extent to which a substance or drug becomes available at its intended biological site of action. Bioavailability is a measure of both the rate and fraction of the administered dose that successfully reaches either the site of action or a bodily fluid where it can exert its effect. Such a fundamental misinterpretation of terminology is unexpected in a manuscript submitted to a reputable journal.
- Corrected as suggested: The correct term is biodiversity, we apologize for the exchange.
- Introduction (Line 79) Molecular Docking: The authors should first justify their selection of molecular targets for docking and explain their relevance to orofacial pain, particularly in the case of iNOS and NMDAR. A brief discussion of the role of iNOS in this context is missing. Additionally, 4-hydroxycoumarin serves as the core structure of anticoagulants, including warfarin and brodifacoum. In lead compound evaluation, MolDock scores are influenced not only by the core structure but also by peripheral moieties. Thus, comparing the binding score of a core scaffold with that of marketed drugs is not scientifically valid.
- Corrected as suggested: The molecular docking analysis revealed no significant interactions with the GABAA, NFκB, µ-opioid, or TRPV receptors, suggesting that these proteins may not be directly involved in the mechanism of action. However, the effects may be synergistic, making in vivo investigation of these targets necessary. In contrast, the complex formed with iNOS exhibited a binding score very close to that of the reference ligand, indicating a potential interaction at this site. Furthermore, the complex with NMDAR demonstrated a higher binding score compared to that of S-ketamine, suggesting a possible inhibitory effect on the NMDAR receptor. These findings imply that the compound may exert its primary effects through interactions with iNOS and NMDAR, rather than the other tested targets. The selection of these molecular targets was based on current literature regarding the mechanisms underlying pain, which involve the activation of various biochemical pathways, including the induction of nitric oxide by iNOS and signaling mediated by NMDAR receptors. The activation of NMDAR receptors allows calcium to enter neuronal cells, which can trigger central sensitization and amplify the transmission of harmful signals. When these receptors are blocked, calcium entry is inhibited, reducing the response to the painful stimulus. In the case of iNOS, its increased expression is associated with heightened inflammation, as the nitric oxide (NO) generated can activate C-fiber nociceptors and contribute to the onset of painful sensations in the spinal cord.
- Corrected as suggested: Regarding the comparison of MolDock scores, we agree with the reviewer that these scores can be influenced not only by the core structure but also by the peripheral moieties of the compounds. However, our data suggest that 4-hydroxycoumarin may be considered a promising starting point for the drug discovery process, based on its interaction with molecular targets related to orofacial pain (such as iNOS and NMDAR). In this context, satisfactory binding interactions and energies were observed, supporting its antinociceptive potential, indicating that this activity may be attributed to properties beyond its pharmacophoric core.
- Moreover, the manuscript transitions directly from in silico experiments to in vivo studies, bypassing mechanistic investigations at the molecular and cellular levels. This research approach lacks a systematic progression and raises concerns about the completeness of the study.
- Corrected as suggested: The direct transition from in silico studies to in vivo assays can be justified by the exploratory and preliminary nature of the research, which focused on the initial validation of the therapeutic activity of 4-HC. The results from the in silico studies provided valuable information that guided the in vivo experiments, allowing for a practical evaluation of the compound's therapeutic viability. We understand that mechanistic investigation is essential for a deeper understanding; however, at the initial stage, we prioritized testing the therapeutic efficacy in a broader context, considering that a more in-depth study of molecular and cellular mechanisms would require additional time and resources. This decision was also driven by the need for a first step toward the functional validation of 4-HC, before delving into the biological details. We believe that, once the initial therapeutic effects are confirmed, more detailed mechanistic studies will be crucial in subsequent phases of the research, providing a more robust understanding of the compound's mechanisms of action.
- Opioid Receptor Experiments The observation in Figure 5B contradicts the conclusion that "the mechanism of action of 4-HC is not through activation of the opioid system" (Line 240). If this conclusion were correct, the effect of 4-HC alone should be comparable to that of 4-HC + naloxone. Furthermore, Lines 236–246 make no mention of morphine, and Figure 5 does not present data for the morphine + naloxone group. This omission is misleading, as it may lead readers to believe that morphine was not included in the experiments. According to my understanding, three experimental groups are involved: 4-HC alone, naloxone + morphine, and naloxone + 4-HC. The naloxone + morphine group should be included in Figure 5. Additionally, the waiting times before the formalin test (Line 495) differ between the administration of morphine and 4-HC. The rationale for this discrepancy should be explained.
- Corrected as suggested: No significant differences were observed between the groups treated with 4-HC (75 mg/kg) combined with naloxone (2 mg/kg) and those that received only hydroxycoumarin. These results, both in the first phase (0-5 min) and in the second phase (15-30 min), suggest that the mechanism of action of 4-HC does not involve the activation of the opioid system (Figures 5A and B). On the other hand, when comparing the group treated with morphine to the morphine + naloxone group, a reversal of the effect is observed, which confirms naloxone’s action and reinforces the mechanism of action of the standard employed in the experiment.
- Corrected as suggested:
Minor Concerns
- The acetic acid-induced writhing test (Figure 1) does not demonstrate dose dependency.
- Corrected as suggested: The acetic acid-induced writhing test does not demonstrate dose dependency as shown in Figure 1.
- The numbering of sections is inconsistent: 2.2, followed by 2.2.1, then 2.3.
- Corrected as suggested: The numbering of the sections follows an organizational logic that reflects the detailed structure of the tests described in the methodology. The subsections within section 2.2 are necessary to clearly and specifically explain the different tests conducted. The transition to section 2.3 occurs when there is a new set of tests, but within the overall context. Thus, the numbering ensures clarity in the presentation of the methodological procedures.
- Line 197 states Phase I: 0–5 min, while Line 239 states Phase I: 0–15 min—this discrepancy should be clarified.
- Corrected as suggested:
- Line 237 should read: "No significant differences were observed..." instead of "No significant results were obtained..."
- Corrected as suggested.
- Line 474, (Sigma Aldrich, Missouri, USA) should be consistent with Line 439, Sigma ( Louis, MO, USA).
- Corrected as suggested.
We appreciate your valuable contributions.

Reviewer 2 Report
Comments and Suggestions for Authors
This manuscript focuses on the in silico and in vivo experiments of 4-hydroxycoumarin. While I appreciate the authors' efforts in preparing the manuscript, several major concerns must be addressed before it can be considered for publication.
Introduction (Line 79) The term "bioavailability" does not mean "available in biology." It refers to the extent to which a substance or drug becomes available at its intended biological site of action. Bioavailability is a measure of both the rate and fraction of the administered dose that successfully reaches either the site of action or a bodily fluid where it can exert its effect. Such a fundamental misinterpretation of terminology is unexpected in a manuscript submitted to a reputable journal.
Molecular Docking The authors should first justify their selection of molecular targets for docking and explain their relevance to orofacial pain, particularly in the case of iNOS and NMDAR. A brief discussion of the role of iNOS in this context is missing. Additionally, 4-hydroxycoumarin serves as the core structure of anticoagulants, including warfarin and brodifacoum. In lead compound evaluation, MolDock scores are influenced not only by the core structure but also by peripheral moieties. Thus, comparing the binding score of a core scaffold with that of marketed drugs is not scientifically valid.
Moreover, the manuscript transitions directly from in silico experiments to in vivo studies, bypassing mechanistic investigations at the molecular and cellular levels. This research approach lacks a systematic progression and raises concerns about the completeness of the study.
Opioid Receptor Experiments The observation in Figure 5B contradicts the conclusion that "the mechanism of action of 4-HC is not through activation of the opioid system" (Line 240). If this conclusion were correct, the effect of 4-HC alone should be comparable to that of 4-HC + naloxone. Furthermore, Lines 236–246 make no mention of morphine, and Figure 5 does not present data for the morphine + naloxone group. This omission is misleading, as it may lead readers to believe that morphine was not included in the experiments. According to my understanding, three experimental groups are involved: 4-HC alone, naloxone + morphine (Line 494), and naloxone + 4-HC. The naloxone + morphine group should be included in Figure 5. Additionally, the waiting times before the formalin test (Line 495) differ between the administration of morphine and 4-HC. The rationale for this discrepancy should be explained.
Minor Concerns
The acetic acid-induced writhing test (Figure 1) does not demonstrate dose dependency.
The numbering of sections is inconsistent: 2.2, followed by 2.2.1, then 2.3.
Line 197 states Phase I: 0–5 min, while Line 239 states Phase I: 0–15 min—this discrepancy should be clarified.
Line 237 should read: "No significant differences were observed..." instead of "No significant results were obtained..."
Line 474, (Sigma Aldrich, Missouri, USA) should be consistent with Line 439, Sigma (St. Louis, MO, USA).
In short, substantial revisions are required before this manuscript can be considered for publication.
Author Response
Dear Editor,
We appreciated the review of our manuscript and the suggestions given to make it suitable to International Journal of Molecular Science high quality standards. As requested, this letter responds to each point raised by the reviewers. Please, note that changes were tracked-up in the manuscript and are transcribed here:
#Reviewer 1:
Minor comments:
- Line 175–181: How does the molecular docking analysis confirm the potential interaction of 4-HC with specific biological targets?
- Corrected as suggested: Molecular docking analysis demonstrated the potential interaction of 4-HC with biological targets by predicting its binding affinity, molecular interactions, and conformational stability. The calculated binding energy indicates the strength of the formed complex, while interactions such as hydrogen bonds and hydrophobic forces reinforce the specificity of the binding. Additionally, the analysis identifies the binding site and suggests a mechanism of action, comparing it with known ligands. These data support further experimental studies, validating the therapeutic potential of 4-HC.
- Line 163–167: What is the significance of the observed binding affinity of 4-HC with NMDAR compared to sketamine?
- Corrected as suggested:
The observed binding affinity of 4-HC with NMDAR suggests that 4-HC forms a more stable complex with this protein compared to sketamine. Once NMDAR is involved in pain signaling, its inhibition could lead to pharmacological implications, disrupting this pathway.
- Line 184–191: How does the acetic acid-induced writhing test validate the antinociceptive properties of 4-HC?
- Corrected as suggested: The acetic acid-induced writhing test is a widely used in vivo experiment to evaluate the antinociceptive properties of compounds. In this assay, intraperitoneal injection of acetic acid causes irritation in the peritoneal cavity, leading to the release of inflammatory mediators such as prostaglandins, cytokines, and nitric oxide, which trigger nociceptive responses characterized by abdominal writhing. A reduction in the number of writhing responses in animals treated with 4-HC, compared to the control group, indicates its ability to suppress pain perception, likely through the inhibition of inflammatory mediators. In this study, 4-HC at doses of 25, 50, and 75 mg/kg showed no significant differences between them. However, the 50 mg/kg dose resulted in the greatest reduction (p<0.05) in the number of abdominal contortions, with a decrease of approximately 36.7% compared to the control group. The 25 mg/kg and 75 mg/kg doses produced respective reductions (p<0.05) of 24.4% and 31.7%. In contrast, the positive control group treated with morphine (5 mg/kg) exhibited a complete abolition of acetic acid-induced abdominal contortions, as shown in Figure 1. These results reinforce the significant peripheral antinociceptive effects of 4-HC, highlighting its potential as an analgesic and anti-inflammatory agent.
- Line 356–360: What evidence suggests that the opioid system is not involved in the antinociceptive effect of 4-HC?
- Corrected as suggested: The pretreatment with naloxone did not antagonize the antinociceptive effect of 4-HC, characterized by the reduction in paw friction, suggesting that the opioid system is not involved in this mechanism. Naloxone, a nonspecific opioid receptor antagonist with high affinity for µ-opioid receptors, acts rapidly; however, its inability to reverse the effect of 4-HC indicates that opioid receptors do not play a significant role in mediating this action. These results are consistent with the in silico study, which showed no affinity (binding and satisfactory energy) for opioid receptors. Together, these findings further support the conclusion that the antinociceptive effects of 4-HC are likely mediated through non-opioid pathways, reinforcing its potential as a novel analgesic with a distinct mechanism of action.
- Line 362–369: How does the carrageenan-induced paw edema model support the anti-inflammatory activity of 4-HC?
- Corrected as suggested: The carrageenan-induced paw edema model is a widely used in vivo test to evaluate anti-inflammatory activity. In this model, the injection of carrageenan into the paw of a rat or mouse induces acute inflammation, characterized by swelling due to the release of pro-inflammatory mediators, such as prostaglandins, cytokines, and histamines. The increase in paw volume serves as an indicator of the inflammatory response. The anti-inflammatory activity of 4-HC is supported by this model if treatment with the compound leads to a significant reduction in the paw edema when compared to the control group. This reduction suggests that 4-HC is able to modulate the inflammatory response, likely by inhibiting the production or activity of pro-inflammatory mediators involved in the edema formation. A significant reduction in paw swelling after 4-HC treatment implies its potential to reduce acute inflammation, which supports its role as an anti-inflammatory agent.
- Line 387–391: What mechanisms are proposed for the reduction of TNF-α levels after 4-HC treatment? Line 94–103: What pharmacokinetic properties of 4-HC suggest its potential as a therapeutic agent?
- Corrected as suggested: The mechanism by which 4-HC lowers TNF-α levels is still undefined. It is proposed that the compound may act through the inhibition of the NFκB pathway via downregulation of pro-inflammatory cytokines. Therefore, additional experimental assays are needed to confirm which pathway is involved in this effect.
- Corrected as suggested: The pharmacokinetic properties of 4-HC, determined through in silico studies, suggest its potential as a therapeutic agent, including good absorption, effective distribution in target tissues, and adequate metabolism and elimination, with no risk of accumulation. Additionally, its low toxicity and wide therapeutic index support its feasibility for preclinical development.
According to pharmacokinetic predictions, 4-HC exhibited favorable properties. The lack of interaction with P-glycoprotein suggests an enhanced safety profile, as this protein functions as an efflux pump, transporting numerous drugs out of cells. Interaction with P-glycoprotein could increase intracellular concentrations of the substance, raising the likelihood of side effects and adverse outcomes [26]. Furthermore, the non-inhibition of CYP P450 enzymes and P-glycoprotein indicates that 4-HC is likely safe and may cause minimal or no drug-drug interactions. CYP enzymes, a subfamily of oxidases, play a central role in the first-pass metabolism of most marketed drugs, and their inhibition can lead to drug interactions and increased adverse effects [27]. Such inhibition could result in changes in pharmacodynamics and elevated toxicological risks [28]. Additionally, 4-HC demonstrated safety in terms of mutagenicity, as evidenced by the Ames test, and carcinogenicity, further supporting its pharmacokinetic safety profile.
- Line 530–531: Why is further research needed to elucidate the pharmacodynamics of 4-HC?
- Corrected as suggested: Therefore, despite the promising initial results, more research is needed to elucidate the pharmacodynamics of 4-HC, as its specific mechanisms of action at the molecular and cellular levels are not fully understood. While pharmacokinetic studies may indicate potential therapeutic benefits, it is crucial to investigate how 4-HC interacts with its biological targets (such as receptors, enzymes, or signaling pathways) to determine its efficacy and safety. Additionally, understanding the dose-response relationship, therapeutic window, and potential side effects will help optimize its therapeutic use. However, this will be addressed in future studies.
#Reviewer 2:
This manuscript focuses on the in silico and in vivo experiments of 4-hydroxycoumarin. While I appreciate the authors' efforts in preparing the manuscript, several major concerns must be addressed before it can be considered for publication.
- Introduction (Line 79) The term "bioavailability" does not mean "available in biology." It refers to the extent to which a substance or drug becomes available at its intended biological site of action. Bioavailability is a measure of both the rate and fraction of the administered dose that successfully reaches either the site of action or a bodily fluid where it can exert its effect. Such a fundamental misinterpretation of terminology is unexpected in a manuscript submitted to a reputable journal.
- Corrected as suggested: The correct term is biodiversity, we apologize for the exchange.
- Introduction (Line 79) Molecular Docking: The authors should first justify their selection of molecular targets for docking and explain their relevance to orofacial pain, particularly in the case of iNOS and NMDAR. A brief discussion of the role of iNOS in this context is missing. Additionally, 4-hydroxycoumarin serves as the core structure of anticoagulants, including warfarin and brodifacoum. In lead compound evaluation, MolDock scores are influenced not only by the core structure but also by peripheral moieties. Thus, comparing the binding score of a core scaffold with that of marketed drugs is not scientifically valid.
- Corrected as suggested: The molecular docking analysis revealed no significant interactions with the GABAA, NFκB, µ-opioid, or TRPV receptors, suggesting that these proteins may not be directly involved in the mechanism of action. However, the effects may be synergistic, making in vivo investigation of these targets necessary. In contrast, the complex formed with iNOS exhibited a binding score very close to that of the reference ligand, indicating a potential interaction at this site. Furthermore, the complex with NMDAR demonstrated a higher binding score compared to that of S-ketamine, suggesting a possible inhibitory effect on the NMDAR receptor. These findings imply that the compound may exert its primary effects through interactions with iNOS and NMDAR, rather than the other tested targets. The selection of these molecular targets was based on current literature regarding the mechanisms underlying pain, which involve the activation of various biochemical pathways, including the induction of nitric oxide by iNOS and signaling mediated by NMDAR receptors. The activation of NMDAR receptors allows calcium to enter neuronal cells, which can trigger central sensitization and amplify the transmission of harmful signals. When these receptors are blocked, calcium entry is inhibited, reducing the response to the painful stimulus. In the case of iNOS, its increased expression is associated with heightened inflammation, as the nitric oxide (NO) generated can activate C-fiber nociceptors and contribute to the onset of painful sensations in the spinal cord.
- Corrected as suggested: Regarding the comparison of MolDock scores, we agree with the reviewer that these scores can be influenced not only by the core structure but also by the peripheral moieties of the compounds. However, our data suggest that 4-hydroxycoumarin may be considered a promising starting point for the drug discovery process, based on its interaction with molecular targets related to orofacial pain (such as iNOS and NMDAR). In this context, satisfactory binding interactions and energies were observed, supporting its antinociceptive potential, indicating that this activity may be attributed to properties beyond its pharmacophoric core.
- Moreover, the manuscript transitions directly from in silico experiments to in vivo studies, bypassing mechanistic investigations at the molecular and cellular levels. This research approach lacks a systematic progression and raises concerns about the completeness of the study.
- Corrected as suggested: The direct transition from in silico studies to in vivo assays can be justified by the exploratory and preliminary nature of the research, which focused on the initial validation of the therapeutic activity of 4-HC. The results from the in silico studies provided valuable information that guided the in vivo experiments, allowing for a practical evaluation of the compound's therapeutic viability. We understand that mechanistic investigation is essential for a deeper understanding; however, at the initial stage, we prioritized testing the therapeutic efficacy in a broader context, considering that a more in-depth study of molecular and cellular mechanisms would require additional time and resources. This decision was also driven by the need for a first step toward the functional validation of 4-HC, before delving into the biological details. We believe that, once the initial therapeutic effects are confirmed, more detailed mechanistic studies will be crucial in subsequent phases of the research, providing a more robust understanding of the compound's mechanisms of action.
- Opioid Receptor Experiments The observation in Figure 5B contradicts the conclusion that "the mechanism of action of 4-HC is not through activation of the opioid system" (Line 240). If this conclusion were correct, the effect of 4-HC alone should be comparable to that of 4-HC + naloxone. Furthermore, Lines 236–246 make no mention of morphine, and Figure 5 does not present data for the morphine + naloxone group. This omission is misleading, as it may lead readers to believe that morphine was not included in the experiments. According to my understanding, three experimental groups are involved: 4-HC alone, naloxone + morphine, and naloxone + 4-HC. The naloxone + morphine group should be included in Figure 5. Additionally, the waiting times before the formalin test (Line 495) differ between the administration of morphine and 4-HC. The rationale for this discrepancy should be explained.
- Corrected as suggested: No significant differences were observed between the groups treated with 4-HC (75 mg/kg) combined with naloxone (2 mg/kg) and those that received only hydroxycoumarin. These results, both in the first phase (0-5 min) and in the second phase (15-30 min), suggest that the mechanism of action of 4-HC does not involve the activation of the opioid system (Figures 5A and B). On the other hand, when comparing the group treated with morphine to the morphine + naloxone group, a reversal of the effect is observed, which confirms naloxone’s action and reinforces the mechanism of action of the standard employed in the experiment.
- Corrected as suggested:
Minor Concerns
- The acetic acid-induced writhing test (Figure 1) does not demonstrate dose dependency.
- Corrected as suggested: The acetic acid-induced writhing test does not demonstrate dose dependency as shown in Figure 1.
- The numbering of sections is inconsistent: 2.2, followed by 2.2.1, then 2.3.
- Corrected as suggested: The numbering of the sections follows an organizational logic that reflects the detailed structure of the tests described in the methodology. The subsections within section 2.2 are necessary to clearly and specifically explain the different tests conducted. The transition to section 2.3 occurs when there is a new set of tests, but within the overall context. Thus, the numbering ensures clarity in the presentation of the methodological procedures.
- Line 197 states Phase I: 0–5 min, while Line 239 states Phase I: 0–15 min—this discrepancy should be clarified.
- Corrected as suggested:
- Line 237 should read: "No significant differences were observed..." instead of "No significant results were obtained..."
- Corrected as suggested.
- Line 474, (Sigma Aldrich, Missouri, USA) should be consistent with Line 439, Sigma ( Louis, MO, USA).
- Corrected as suggested.
We appreciate your valuable contributions.
Round 2
Reviewer 2 Report
Comments and Suggestions for Authors
I appreciate the author's effort in revising the manuscript.